# Tracking of Melanoma Cell Plasticity by Transcriptional Reporters

**DOI:** 10.3390/ijms23031199

**Published:** 2022-01-21

**Authors:** Anna Vidal, Torben Redmer

**Affiliations:** 1Institute of Medical Biochemistry, University of Veterinary Medicine, 1210 Vienna, Austria; Anna.Vidal@vetmeduni.ac.at; 2Unit of Laboratory Animal Pathology, Institute of Pathology, University of Veterinary Medicine, 1210 Vienna, Austria

**Keywords:** transcriptional reporters, plasticity, stemness, NGFR

## Abstract

Clonal evolution and cellular plasticity are the genetic and non-genetic driving forces of tumor heterogeneity, which in turn determine tumor cell responses towards therapeutic drugs. Several lines of evidence suggest that therapeutic interventions foster the selection of drug-resistant neural crest stem-like cells (NCSCs) that establish minimal residual disease (MRD) in melanoma. Here, we establish a dual-reporter system, enabling the tracking of NGFR expression and mRNA stability and providing insights into the maintenance of NCSC states. We observed that a transcriptional reporter that contained a 1-kilobase fragment of the human NGFR promoter was activated only in a minor subset (0.72 ± 0.49%, range 0.3–1.5), and ~2–4% of A375 melanoma cells revealed stable NGFR mRNA. The combination of both reporters provides insights into phenotype switching and reveals that both cellular subsets gave rise to cellular heterogeneity. Moreover, whole transcriptome profiling and gene-set enrichment analysis (GSEA) of the minor cellular subset revealed hypoxia-associated programs that might serve as potential drivers of an in vitro switching of NGFR-associated phenotypes and relapse of post-BRAF inhibitor-treated tumors. Concordantly, we observed that the minor cellular subset increased in response to dabrafenib over time. In summary, our reporter-based approach provides insights into plasticity and identified a cellular subset that might be responsible for the establishment of MRD in melanoma.

## 1. Introduction

The emergence of therapy-resistant tumor cell clones is frequently observed in melanoma patients and is responsible for tumor relapse and poor prognosis. Recently, a neural crest stem cell (NCSC)-like signature has been identified, which determines minimal residual disease in melanoma [1]. Hence, high levels of NCSC genes likely confer intrinsic resistance to the inhibition of BRAF and MEK, NGFR being among them. The expression of the latter nerve growth factor receptor (CD271) and putative marker of melanoma initiating cells sufficiently mediated resistance to vemurafenib in vitro and was associated with lymph node metastasis in melanoma patients [2,3,4,5,6,7]. Melanoma cells expressing NGFR exhibit increased migratory, invasive and survival capacities and the knockdown of this TNF-receptor family member revealed a network of NGFR/CD271-associated genes that facilitate and maintain different phenotypes [8,9,10,11,12]. Although NGFR-mediated signaling mechanisms are still not well understood in melanoma and cancer, several lines of evidence suggest that NGFR serves as a regulator of basic tumor cell properties such as plasticity, hence, non-genetic and reversible switching of cellular phenotypes [13,14,15,16] likely triggered by environmental cues causing stress. Concordantly, NGFR was highly expressed in chemoresistant melanoma cells [10].

Here, we present our established dual-reporter system, which enables the tracking of NGFR expression over time and demonstrates the phenotype-switching capacity of isolated cellular subsets in response to therapeutic drugs.

## 2. Results and Discussion

### 2.1. A Dual NGFR Reporter Revealed Phenotype Switching of Melanoma Cells

Melanoma cells feature a high capacity of cellular plasticity, a non-genetic process that likely enables adaptation to environmental cues such as changes in growth factor conditions or interaction with extracellular matrix components [16]. In addition, cellular plasticity probably controls the interconversion of stem-like and non-stem-like tumor cells. The expression of NGFR/CD271 has been associated with melanoma cell stemness [3,11,12,17]; however, we observed that the CD271^+^ population contains a label-retaining subset and, hence, comprises a proliferating non-stem-like fraction and a slow-cycling fraction, which likely presents the actual subset of melanoma-initiating cells [11,12]. These previous findings suggest that CD271 labels a heterogeneous pool of melanoma cells, which likely feature unique and common properties. We established a double-reporter system that combines the transcriptional activation of a 1-kb fragment of the NGFR promoter that controls the expression of RFP and of a sequence of the 3′-UTR of NGFR that controls the stability of GFP, expressed under the control of a cytomegalovirus (CMV) promoter. The NGFR promoter sequence comprised a region of 900 bp upstream to 100 bp downstream from the transcriptional start site (Appendix A) and contains binding sites for early growth response protein 1 (EGR1). The latter, in turn, has been associated with the control of NGFR expression [18]. Hence, the reporters reflected two states of NGFR expression: the transcriptional activation of the NGFR promoter and the regulation of mRNA stability (Figure 1A, left schemes). The sequential lentiviral transduction of A375 cells revealed the co-existence of four discrete cell states: GFP^+^/RFP^neg^, GFP^+^/RFP^+^, RFP^+^/GFP^neg^ and GFP^neg^/RFP^neg^ (Figure 1B, right panel). Next, we asked whether the different cell states that were investigated ~14d post infection and selection for puromycin resistance presented stable or interconvertible states and, thus, reflected phenotype switching. To this end, we analyzed the levels of GFP and RFP 7 d after a FACS-based isolation of subsets presenting the four cell states, and we hypothesized that the double-positive subset constituted an intermediate state. We investigated the occurrence of GFP^+^, RFP^+^ and double-positive subsets and observed that all four cell states emerged from the different isolated subsets (Figure 1B,C). Moreover, we observed that GFP^+^ cells showed lower cell surface expressions of CD271 than RFP^+^ cells, suggesting that the transcriptional reporter superiorly reflected the levels of CD271 and that the expression of GFP indeed recapitulated the post-transcriptional regulation. Although FACS-based enrichment was highly efficient, we could not fully discount that the subsets might have contained admixed cells of other subsets. Therefore, we investigated the emergence of different cell states from double-negative subsets. We observed that all four cell states emerged from double-negative cells over time (Appendix A) and maintained the expression of melanoma marker KBA.62. In addition, we observed that even double-positive cells were capable of developing double-negative cells (Figure 1C, lower row). Double-positive cells likely presented an intermediate population that featured a multipotent, stem-like capacity. However, intrinsic programs might control the capacity of renewing the NGFR^+^ cell pool independent from the initial phenotypes. Therefore, the reporters recapitulated the plasticity of NGFR/CD271^+^ phenotypes of in vitro cultured cells, as reported in previous studies [19,20,21]. However, the in vivo stability and sustainability of the NGFR/CD271^+^ phenotype of melanoma cells are unknown and possibly inaccessible for investigation.

We validated the functionality of the transcriptional reporter in A2058 cells (Appendix A); however, we observed a high reliability of the transcriptional reporter only in A375 cells. Generally, we not only observed a high concordance of reporter activation and cell surface levels of CD271 but also found minor subsets that featured activation of the transcriptional reporter (RFP^+^) whilst showing only a low level of CD271 (Figure 1D, left panels). This discrepancy might reflect different regulatory circuits controlling the level of CD271 expression. In summary, the NGFR double-reporter system enabled the tracking of phenotype switching and demonstrated the multipotent capacity of melanoma cells (Figure 1D, right panel), although low-passage, patient-derived melanoma cells might feature a different turnover of cellular phenotypes.

### 2.2. Whole Transcriptome Profiling Revealed a Hypoxic Phenotype of Cellular Subsets

RFP^+^ cells sufficiently renewed the stem-like cell pool by symmetric cell division (Figure 2A) and gave rise to all subsets. Therefore, we asked which molecular programs activated in the minor cellular subset potentially featured activation of the transcriptional reporter. We determined the proportion of proliferative/Ki67^+^ cells within the RFP^+^ subset, and we observed a high percentage of RFP^+^/Ki67^+^ cells using flow cytometry (89.6%) and quantitative immunofluorescence microscopy (54.2 ± 7.6%) (Figure 2B). Next, we performed whole transcriptome profiling of FACS-isolated RFP^+^ and unsorted bulk cells and identified 607 differentially regulated genes (*p* < 0.001), among them the ABC transporter ABCC2, MMP1 (matrix metalloproteinase 1), IL24 (interleukin 24) and the hypoxia-induced TXNIP (thioredoxin interacting protein) (Figure 2C, left panel). Concordantly, gene-set enrichment analysis (GSEA) revealed a significant enrichment of signature genes of hypoxia, EMT (epithelial-to-mesenchymal transition) and TNFα-induced NFκB signaling (Figure 2C, right panel). In addition, we performed single-sample GSEA and observed the separation of RFP^+^ and bulk cells using signatures that are associated with hypoxia, stemness and NFκB signaling (Figure 2D). The expression of CD271 is linked to neural crest cell stemness in melanoma and the downregulation of NGFR/CD271 abrogated all features of a neural crest stem cell (NCSC)-like state [11]. We asked whether RFP^+^ cells featured an NCSC-like or intermediate cell state. Indeed, GSEA revealed a proliferative and transitory-melanocytic rather than an NCSC state (Figure 2E). This finding underpinned the heterogeneity of the CD271^+^ cell pool, which comprised proliferating and non-proliferating cells. However, we observed that only a very rare subset of cells indeed exhibited a slow-cycling phenotype, and the majority of cells lost the labeling by the lipophilic dye PKH67 over time. In addition, the label-retaining capacity of the RFP^+^ subset and bulk cells was comparable (Appendix A).

### 2.3. NGFR/RFP^+^ Cells Are Enriched upon Dabrafenib Treatment Reflecting a MAPKi-Driven Hypoxic-Phenotype

Several lines of evidence suggest a therapy-driven enrichment of intrinsically resistant stem-like melanoma cells establishing minimal residual disease (MRD) as a driving force of relapse in melanoma patients. We investigated pre- and post-treatment melanoma specimens of patients who received BRAFi/MEKi (dabrafenib/trametinib) therapy and developed therapy-resistant progressive disease (GSE7794027 [22]). The expression of NGFR but not of ABCB5 (an additional marker of melanoma stem-like cells [23]) was increased in five of six post-treatment tumors (*p* = 4 × 10^−3^; two-sided paired *t*-test) and was accompanied by decreased levels of E-cadherin/CDH1 (Figure 3A, left panel). Moreover, GSEA revealed an enrichment of hypoxia-related genes, hence suggesting the activation of a hypoxia-related program in relapsed tumors (Figure 3A, right panel).

We asked whether the minor cellular subset, as marked by activity of the transcriptional reporter, presented an MRD-establishing cell clone. As we expected that the clone featured resistance towards dabrafenib, we performed a live-cell imaging-based tracking of A375^CD271-RFP^ reporter cells for 72–96 h and analyzed changes in the proportion of RFP^+^ cells over time. Reporter cells showed a general response towards dabrafenib in a range of 1–3 nM (IC50 = 2.148 nM) and a dose-dependent increase in the number of RFP^+^ cells (Figure 3B and Appendix A) in a range according to IC50. The dabrafenib response of reporter cells was evident even 48 h after treatment, which reached a maximum after 72–92 h (Figure 3C and Appendix A) and was not accompanied by changes in the number of GFP^+^ cells (data not shown). However, this increase very likely reflected the dabrafenib-induced activation of the reporter and potentially marked melanoma cells that featured MRD. Next, we assessed whether gene signatures mirroring an in vivo drug response and potentially reflected MRD-states were enriched in the RFP^+^ minor subset. Although we observed a partial association by GSEA, none of the MRD-related gene signatures showed a significant enrichment in the RFP^+^ subset (data not shown).

As hypoxia serves as a driving force of therapy resistance [24], we asked whether the hypoxic phenotype of RFP^+^ cells might aid cells escaping from a dabrafenib-mediated downregulation of proliferative programs. We observed that the hypoxia-inducible transcription factors EGR1 and ATF3 were significantly higher expressed in RFP^+^ than in bulk cells (Figure 3D, left panel, Appendix A) and likely mediated reporter activation in response to dabrafenib by binding to their respective sites in the NGFR promoter. Hence, the hypoxia-driven expression of both transcription factors might be responsible for the increased expression of CD271/NGFR in response to dabrafenib.

Indeed, ATF3 and NGFR were the most enriched in vemurafenib-treated COLO858 cells [25], and EGR1 expression was significantly increased in the CD271/NGFR^high^ subset of melanoma (TCGA data) (Figure 3D, right panel). The binding of EGR1 to the NGFR promoter has been previously shown [18], suggesting that EGR1 might act in concert with ATF3 and tightly controls the expression of CD271 during phenotype switching. Finally, we investigated the potential regulation of general markers of melanoma (PD-L1, MET) and of additional mediators of stemness and phenotype switching in addition to NGFR, such as SOX9, ATF3 and TMEM47 [26,27,28,29,30,31,32] in RFP-positive (RFP) or RFP-negative (−) sorted cells four and eight days after sorting. We observed increased expression levels from day four to eight; ATF3 and NGFR, in particular, revealed a steady increase in expression levels (Appendix A). Therefore, additional markers are likely regulated by phenotype switching. In particular, the regulation of PD-L1, the target of immune checkpoint inhibitor-based therapies, may determine the efficacy of therapeutic interventions targeting a certain cell surface molecule [33].

In summary, we have established an NGFR reporter system that enabled the tracking of phenotype switching and drug response and uncovered a potential role for hypoxia and hypoxia-induced gene expression in the emergence of progressive dabrafenib-resistant melanoma.

## 3. Materials and Methods

### 3.1. Cell Culture

A375 and A2058 (purchased from ATCC) were kept at 37 °C, 5% CO_2_ and 95% humidity in cell culture medium (DMEM, 4.5 g/L glucose, stabilized glutamine/GlutaMAX, pyruvate) supplemented with 10% fetal bovine (GIBCO/Thermo Fisher, Waltham, MA, USA) serum and 1% penicillin/streptomycin (GIBCO/Thermo Fisher) as previously reported [11]. Cells were seeded onto 8-chamber glass slides to a density of 5000–10,000 cells per chamber for imaging. Label-retention assays were performed as previously described [11].

### 3.2. Flow Cytometry/Fluorescence-Activated Cell Sorting (FACS)

After the removal of medium, cells were washed with 1X phosphate-buffered saline (PBS) and harvested by trypsin (0.05% trypsin/EDTA). Following the addition of cell culture medium, cells were collected by centrifugation at 1000 rpm at room temperature for 3 min (min) and resuspended in 100 µL of ice-cold buffer (1X PBS/0.5% bovine serum/2 mM EDTA) and stored on ice. Cells were incubated with primary antibodies CD271-APC (Miltenyi, Bergisch Gladbach, Germany), RFP (Novus Biologicals, Abingdon, United Kingdom) or KBA.62 (BioLegend, San Diego, CA, USA), all diluted to 1:80 in buffer and stored at 4 °C for 10 min to achieve proper labeling. Following this, cells were washed with buffer and incubated with secondary antibodies (Alexa Fluor-488/594/647, diluted to 1:500) and incubated as mentioned above. Washed cells were resuspended in 500 µL PBS and analyzed by flow cytometry (Canto II; Becton, Dickinson and Company, Franklin Lakes, NJ, USA). The isolation of reporter cells was performed with Aria III (Becton, Dickinson and Company, Franklin Lakes, NJ, USA) regarding their levels of GFP and RFP, collected in cell culture medium and expanded under normal cell culture conditions. Data analysis was performed with FlowJo, (https://www.flowjo.com/solutions/flowjo/downloads (accessed on 20 December 2021); Ver 10.7.1). FACS-isolated cells were collected in cell culture medium and seeded on appropriate vessels following centrifugation.

### 3.3. Production of Lentiviral Particles

Briefly, for the production of lentiviral particles, Lenti-X cells were seeded to 1 × 10^4^ cells on a 10-centimeter dish and transfected after 24 h with 4 µg of a plasmid expressing GFP fused to the 3’-untranslated region (UTR) of NGFR (abmGood, #3180008) to elucidate interactions between miRNAs and the 3′UTR of the NGFR gene or a customized plasmid that contained a 1-kilobase (kb) promoter (−900; +100) of the human NGFR gene and the coding sequence of red fluorescent protein (RFP). The RFP reporter plasmid was customized by Applied Biological Materials Inc. (ABM, Richmond, BC, Canada). For generation of viral particles, Lenti-X cells were transfected with 2 µg of pMD2.G (Addgene # 12259, VSV-G envelope) and 1 µg of psPAX2 (# 12260) packaging plasmids using 20 µL/1 mL PEI (polyethylenimine, Sigma-Aldrich, St. Louis, MO, USA). The medium was changed after 24 h and viral supernatant was harvested after an additional 24 h. Viral supernatants were filtered through a 0.45-micrometre filter and applied to target cells for 24–48 h. Infected cells were selected for resistance to puromycin (10 µg/mL).

### 3.4. Live-Cell Imaging-Based Drug Assays

The response of A375 cells to dabrafenib in a range of 1 nM–10 µM of eight technical replicates was assessed by live-cell imaging-based drug responses using an IncuCyte S3 live-cell imaging microscope (Sartorius, Goettingen, Germany) 24 h after seeding of 2500 cells/100 µL/96-well and the addition of serially diluted compounds. Images were taken every three hours using a 10× objective and the general label-free mode. Two pictures of eight technical replicates per condition were taken. Drug response was assessed by changes in cellular density over time. The cell density was determined by a confluence mask tool as part of the IncuCyte software, Ver2021A. IC50 values were calculated by curve-fitting (https://search.r-project.org/CRAN/refmans/REAT/html/curvefit.html; accessed on 12 September 2021) based on confluence measurements at day 3 and IncuCyte software tools.

### 3.5. Immunofluorescence and Confocal Microscopy

Briefly, following the removal of medium cells grown in chamber, slides were washed once with PBS and fixed with PFA (paraformaldehyde, 4% in PBS) for 10 min at room temperature. After fixation, washed cells were incubated with primary antibodies p75NTR (1:100; Cell Signaling Technology, Danvers, MA, USA) and RFP (Novus Biologicals, 1:200) were diluted in blocking buffer (PBS, 2% fetal serum albumin) overnight at 4 °C. On the next day, cells were washed and incubated with secondary antibodies (Alexa Fluor-488/594/647, diluted to 1:500) and DAPI (4′,6-diamidino-2-phenylindole, diluted to 1:500) and incubated for 1 h at room temperature and protected from light. Following this, slides were covered with mounting medium and cover slips and stored at 4 °C until imaging.

High-resolution immunofluorescent imaging of tumor sections and cell lines was performed with an LSM 880 Airyscan confocal microscope (Carl Zeiss AG, Oberkochen, Germany) and appropriate software (Zen Black, ver. 2.3 SP1, Carl Zeiss AGs). Images were taken with objectives 10×, 20× and 63×/1.40 (plan-apochromat, oil DIC M27) at a resolution of 2048 × 2048 pixels/cm, 8bit, scan speed 6, averaging 4. Immersol 518F was used for oil microscopy. Stacked multichannel image files (CZI) were separated and background adjusted using Adobe Photoshop 2020 (Adobe, San Jose, CA, USA) and stored as merged TIFF files at a resolution of 600 dpi.

### 3.6. RNA Isolation and Sequencing

Briefly, isolation of total RNA of snap frozen or intraoperative tumors was performed with the RNAeasy extraction kit (Quiagen, Germantown, MD, USA) according to the manufacturer’s instructions. RNA integrity was determined by automated electrophoresis (4200 TapeStation system (Agilent, Santa Clara, CA, USA). The library preparation of 100 ng total RNA was performed with TruSeq Stranded Total RNA Sample Preparation-Kit and Ribo Zero Gold (Illumina, San Diego, CA, USA), and paired-end (2 × 100 bp) whole transcriptome profiling of RNA with integrity numbers (RIN) ≥ 7 was performed at CeGaT GmbH (Tübingen, Germany) and sequenced on the NovaSeq 6000 platform. Illumina bcl2fastq (2.19) was used for a demultiplexing of sequenced reads, and adapter trimming was performed with Skewer (version 0.2.2) [34]. The information on FASTQ files was obtained by using the FastQC program (version 0.11.5-cegat) read out. Raw sequencing data (FASTQ files) were quality controlled using FastQC (version 0.11.7—Bioinformatics Group at the Babraham Institute) and further preprocessed with FASTP [35]. Reads were aligned to the GRCh38 version of the human genome using TopHat [36], and counts per gene were calculated by the featureCount algorithm from the Rsubread package [37]. All further steps of the analysis were performed in R. Raw counts of protein-coding genes were normalized using the DESeq2 (https://bioconductor.org/packages/release/bioc/html/DESeq2.html; accessed on 12 February 2019) package [38]. Differential expression of genes between groups was determined after fitting models of negative binomial distributions to the raw counts. Raw *p*-values were fdr-adjusted for multiple testing, and a value below 0.05 for the adjusted *p*-values was used to determine significant differentially expressed genes. Functional annotation of genes, over representation and gene-set enrichment analysis were performed using the clusterProfiler package [39]. For visualization of differentially expressed genes and molecular subgroups, we used EnhancedVolcano (https://bioconductor.org/packages/EnhancedVolcano/; accessed on 20 December 2021).

### 3.7. Gene-Set Enrichment GSEA/Single-Sample GSEA

GSEA was performed by using the most current BROAD java GSEA standalone version (http://www.broadinstitute.org/gsea/downloads.jsp; accessed on 10 December 2021) and gene signatures of the molecular signature database MSigDB [17,40], 7.4 (Hallmark, C2), as well as published signatures specifying different phenotypic states of melanoma such as gene signatures defining the “transitory-melanocytic” or proliferative state that were taken from Tsoi et al. [41] or Verfaillie et al. [42]. Analyses of single signatures were run using 10,000 permutations; analyses of signature collections were run using 1000 permutations. Genes were ranked based on the Signal2Noise metric.

### 3.8. Quantitative Real-Time PCR

RNA isolation from frozen cell pellets was performed with the RNeasy Mini Kit (Quiagen, Germantown, MD, USA) and by following the manufacturer’s protocol. Reverse transcription of 500 ng–2.5 µg RNA was performed with SuperScript VILO cDNA synthesis kit (Thermo Fisher, Waltham, MA, USA) and diluted to a final volume of 50 µL. qRT-PCR was carried out on a StepOnePlus (Applied Biosystems, Waltham, MA, USA) for 30–40 cycles. Primers were designed for 55–60 °C annealing temperatures. Relative expression levels were calculated with ΔΔCT and normalized to β-actin. Primer sequences are shown in the Appendix A.

## Figures and Tables

**Figure 1 ijms-23-01199-f001:**
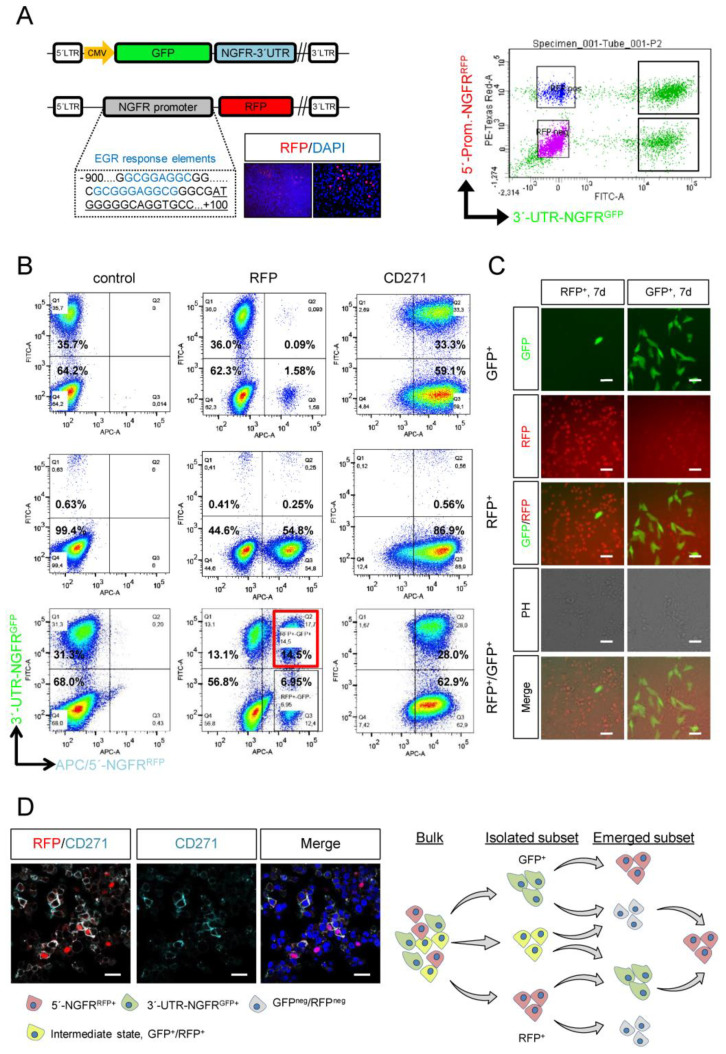
A dual−reporter system enabled tracking of phenotype switching. (**A**) Simplified plasmid maps of 3′-UTR-NGFR miRNA reporter that expressed GFP fused to 3′-UTR of the NGFR gene under control of a CMV promoter (upper map) and of a customized NGFR reporter that expressed RFP under control of a 1-kilobase fragment of the NGFR promoter that contained EGR response elements (lower map). Images depict indirect detection of RFP^+^ cells using an RFP-specific antibody. A FACS plot indicates the presence of four distinct cellular subsets regarding the expression of reporters that were stably expressed in A375 cells (right panel). (**B**) Flow cytometric analysis of FACS-isolated cellular subsets GFP^+^, RFP^+^ or double-positive (GFP^+^/RFP^+^) 7 d post isolation for levels of GFP, RFP and cell surface expression of CD271; 50,000–100,000 cells were recorded. (**C**) Immunofluorescence microscopy of A375 reporter cells analyzed in (**B**) showing the distribution of cellular subsets. Bars indicate 50 µm. (**D**) Confocal microscopy of reporter cells stained for RFP and CD271 indicating distinctness and co-occurrence of both. Bars indicate 50 µm; DAPI served as nuclear stain (left panels). Schematic representation of phenotype-switching processes as tracked by the dual-reporter system.

**Figure 2 ijms-23-01199-f002:**
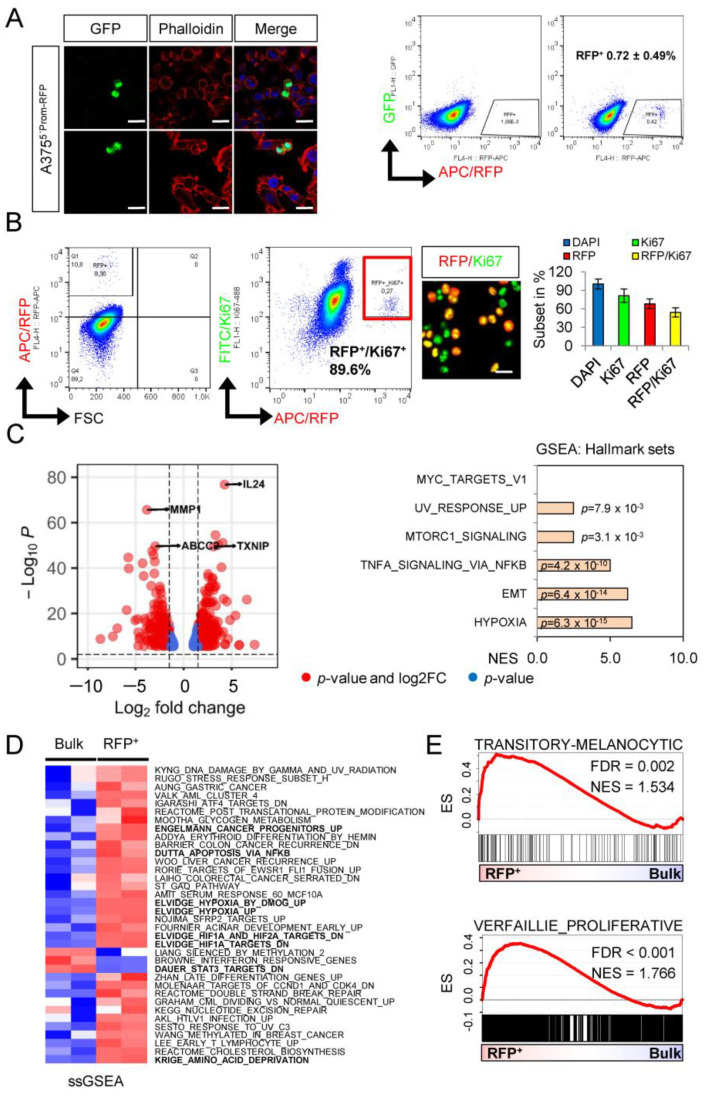
A minor cellular subset featured a hypoxic and proliferating phenotype. (**A**) Confocal microscopy of reporter cells revealed a symmetric division of RFP^+^ cells (left panels) and flow cytometry of 10,000 recorded cells indicated a low abundance of A375 cells showing activation of the transcriptional reporter. (**B**) Flow cytometric analysis of RFP^+^ cells revealed a high percentage of proliferating Ki67^+^ cells within the RFP^+^ cell pool (left panels). Concordantly, the content of Ki67^+^ cells was increased in the RFP^+^-enriched subset (right panels). (**C**) Volcano plot depicts the top differentially regulated genes (DEGs, *p* < 0.001) as identified by whole transcriptome profiling of RFP^+^ cells, showing that IL24 and TXNIP, and MMP1 and ABCC2, were among the top up-regulated and down-regulated genes, respectively (left panel). GSEA of DEGs revealed enriched hallmark signatures with hypoxia and EMT as most significantly enriched pathways in RFP^+^ cells (right panel). (**D**) Single-sample GSEA revealed that hypoxia-associated gene signature among others significantly (*p* < 0.001) separated RFP^+^ from bulk cells. (**E**) Enrichment plots indicate the predominance of genes that are associated with a transitory melanocyte or proliferative cell phenotype.

**Figure 3 ijms-23-01199-f003:**
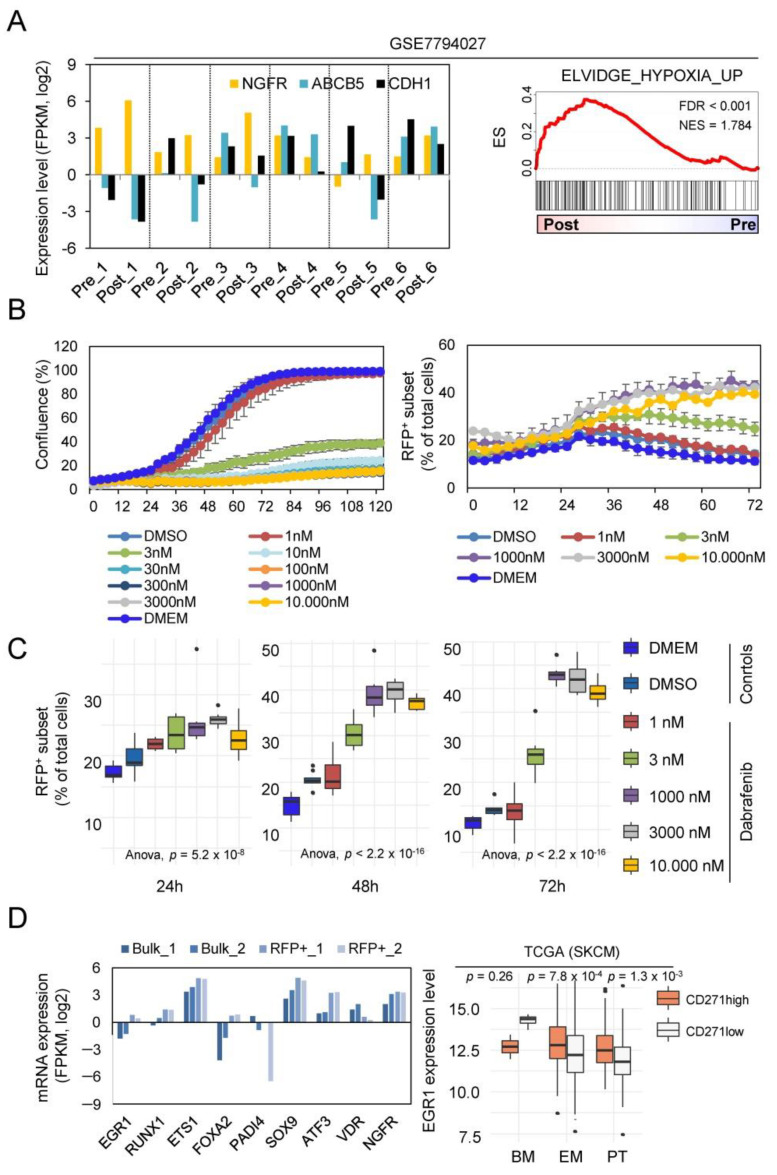
Dabrafenib treatment increased the emergence of resistant cell clones. (**A**) FPKM (log2) expression levels of markers of a melanoma stem-like state NGFR and ABCB5 and the epithelial marker E-cadherin (CDH1) in six pre-BRAFi treatment and post-relapse patient-derived tumors of melanoma origin (GSE7794027) (left panel). GSEA of public data analyzed in (**A**) for enrichment of hypoxia-associated genes shows enrichment of hypoxia-related genes in the group of therapy-resistant, relapsed tumors (right panel). ES = enrichment score; NES = normalized enrichment score. (**B**) Live-cell imaging-based dose–response of reporter cells towards dabrafenib. Drug response is indicated by changes in confluence (%), left panel. Increase in the number of RFP^+^ cells as depicted in % of total cells of dabrafenib treated cells (right panel). Values are shown as mean ± SDV of eight technical replicates; a representative out of three experiments is shown. (**C**) Box plot representation of cell density at 24, 48 and 72 h after treatment depicts a significant difference in the number of RFP^+^ cells, depicted as percentage of total cells (right panel). (**D**) Expression levels of top differentially regulated genes among RFP^+^ and bulk cells; log2 FPKM values of two independent biological replicates are shown (left panel). Box plot representation of levels of EGR1 expression in the set of TCGA melanoma (PT, primary melanoma; EM, extracranial; BM, brain metastases) ranked by expression of CD271/NGFR; significance was determined by a two-way ANOVA.

## Data Availability

In this study we performed analysis of public data (GSE7794027) as well as on the TCGA melanoma data set (SKCM); https://portal.gdc.cancer.gov/projects/TCGA-SKCM (accessed on 20 December 2021). Appendix A can be downloaded from https://doi.org/10.5281/zenodo.5840821 (accessed on 20 December 2021).

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
