# Peer review of "Tracking of Melanoma Cell Plasticity by Transcriptional Reporters"

_ijms, 2022, doi:10.3390/ijms23031199_

Round 1

Reviewer 1 Report

This is a very interesting and valuable contribution where it is established a dual-reporter system that enabled the tracking of NGFR (nerve growth factor receptor) expression over time in melanoma cell lines. It is also demonstrated the phenotype switching capacity of isolated cellular subsets in response to therapeutic drugs, that is probably related with the chemoresistance and metastatic potential of this sort of cancer. The paper clearly shows that the downregulation of NGFR/CD271 (a novel NGFR receptor) abrogated all features of a neural crest stem cell (NCSC)-like state.

Introduction is well focused, methods are smart and appropriate, including all checked links, and results and discussion are consistent, well balanced and organized. References are also appropriated.

The following minor points should be addressed before definitive acceptance

Lines 50-51: “Melanoma cells feature a high capacity of cellular plasticity, a non-genetic process 50 that likely enables the adaptation to environmental cues”

This statement would be somehow corrected. As authors undoubtedly known, melanoma cells are malignant melanocytes, and melanocytes are derived from the neural crest and share with neurons genetic factors to form dendritic branches for plasticity. Give some examples of the environmental cues would be informative.

Figures have excellent quality, so that they are clear and have informative legends. However, Figure S1 (A, B, C and D panels) needs some legend. As far as I can see, Fig S1A showing a DNA sequence is never mentioned throughout the text, and it should be. As a suggestion, authors would reconsider the use of this Figure S1 as Supplementary figure instead of being incorporated to the paper as one more to facilitate the reading of the manuscript to future readers.

Lines 207 and 209, Ref. 11 should be cited as a superscript.

Author Response

Letter of response

Dear Reviewer #1,

thank you very much for your time and provision of constructive comments and suggestions. Please find our point-by-point response below. All parts that have been changed at indicated places in the manuscript are highlighted in yellow.

This is a very interesting and valuable contribution where it is established a dual-reporter system that enabled the tracking of NGFR (nerve growth factor receptor) expression over time in melanoma cell lines. It is also demonstrated the phenotype switching capacity of isolated cellular subsets in response to therapeutic drugs, that is probably related with the chemoresistance and metastatic potential of this sort of cancer. The paper clearly shows that the downregulation of NGFR/CD271 (a novel NGFR receptor) abrogated all features of a neural crest stem cell (NCSC)-like state.

Introduction is well focused, methods are smart and appropriate, including all checked links, and results and discussion are consistent, well balanced and organized. References are also appropriated.

The following minor points should be addressed before definitive acceptance

Lines 50-51: “Melanoma cells feature a high capacity of cellular plasticity, a non-genetic process 50 that likely enables the adaptation to environmental cues”

This statement would be somehow corrected. As authors undoubtedly known, melanoma cells are malignant melanocytes, and melanocytes are derived from the neural crest and share with neurons genetic factors to form dendritic branches for plasticity. Give some examples of the environmental cues would be informative.

Response: We agree and have now included some examples and a reference stating that particularly changes in the supply of nutrients and microenvironment make a phenotype switching necessary (page 2, lines 52-53). 

Figures have excellent quality, so that they are clear and have informative legends. However, Figure S1 (A, B, C and D panels) needs some legend. As far as I can see, Fig S1A showing a DNA sequence is never mentioned throughout the text, and it should be.

Response: We absolutely agree and have now prepared a separate file containing all supplementary figures and figure legends. Figure S1A is now linked with main text (page 2, line 65).

As a suggestion, authors would reconsider the use of this Figure S1 as Supplementary figure instead of being incorporated to the paper as one more to facilitate the reading of the manuscript to future readers.

Response: Thank you very much for this suggestion, however we think that the incorporation of the supplementary data would be difficult.

Lines 207 and 209, Ref. 11 should be cited as a superscript.

Response: We agree and corrected this.

General comments: Regarding the concerns of reviewer #2 and our own concerns we now included an additional live cell-imaging based dose-response experiment that demonstrates a response of A375 cells towards dabrafenib at a dosis of ~ 2 nM. This low dose response was accompanied by an increase in the number of RFP+ cells (page 3, lines 142-149).  In addition, we investigated publicly available data of pre-BRAFi and post-treatment, relapsed melanoma for levels of NGFR and ABCB5. Although this data set comprised only a set of six patients, we observed that the expression of NGFR but not ABCB5 was indeed increased in five of six post-treatment melanoma (page 3, lines 133-138). As GSEA of same samples revealed enrichment of hypoxia-related genes in post-treatment melanoma, we suggest that our reporter system probably recapitulates the response observed in tumors. However, we do not have mechanistic insights yet.   

Reviewer 2 Report

The authors address a well known issue in melanoma research -- cell plasticity, using molecular labeling and tracking. The data are intriguing, but not unexpected based on previous studies in the field. Figure 3A requires additional contrast in order to visualize the data. In general, the study could be strengthened by adding an additional well characterized marker for stemness and by incorporating functional data to correlate with the phenotype switching. This would more clearly show a cause-and-effect relationship associated with the key elements of this study.

Author Response

Letter of response

Dear Reviewer #2,

thank you very much for your time and provision of constructive comments and suggestions. Please find our point-by-point response below. All parts that have been changed at indicated places in the manuscript are highlighted in yellow.

The authors address a well known issue in melanoma research -- cell plasticity, using molecular labeling and tracking. The data are intriguing, but not unexpected based on previous studies in the field. Figure 3A requires additional contrast in order to visualize the data. In general, the study could be strengthened by adding an additional well characterized marker for stemness and by incorporating functional data to correlate with the phenotype switching. This would more clearly show a cause-and-effect relationship associated with the key elements of this study.

Response: We absolutely agree that mechanic insights would strengthen the manuscript. However, this is a proof-of-principle study showing that our reporters represent phenotypic cell states and in principle responded to dabrafenib treatment. Therefore, the reporter system might be suitable for in vitro and in vivo tracking of resistant cell states and can be combined with additional reporters. Moreover, the system might enable the identification of drugs that trigger the formation of a NGFR-driven stem-like state on one hand and potentially facilitates a screening for stemness-targeting small molecule drugs.

We also investigated effects of paclitaxel and tivantinib (ARQ197), an inhibitor of c-MET signaling, however we did not observe increased numbers of RFP+ cells under these conditions.

Regarding your and our own concerns, we now included an additional live cell-imaging based dose-response experiment that demonstrates a response of A375 cells towards dabrafenib at a dosis of ~ 2 nM. This low dose response was accompanied by an increase in the number of RFP+ cells (page 3, lines 142-149).  In addition, we investigated publicly available data of pre-BRAFi and post-treatment, relapsed melanoma for levels of NGFR and ABCB5. Although this data set comprised only a set of six patients, we observed that the expression of NGFR but not ABCB5 was indeed increased in five of six post-treatment melanoma (Figure 3A, page 3, lines 133-138). As GSEA of same samples revealed enrichment of hypoxia-related genes in post-treatment melanoma, we suggest that our reporter system probably recapitulates the response observed in tumors. However, we do not have mechanistic insights yet. Besides NGFR, we investigated additional markers that we identified by RNAseq, however the well-known markers of stemness ABCB5 or PROM1 were not expressed and hence not among the genes that were found differentially regulated (DEGs) between bulk and RFP+ cells. Among DEGs, we identified SOX9 and ATF3, putative mediators of phenotype switching and stemness. The data are included in a new supplementary figure (Figure S2) and described as mentioned above.

Round 2

Reviewer 2 Report

The authors have done an excellent job revising this manuscript.  If possible, additional or enhanced contrast to Sup. Fig. 1A would improve the RFP data.